# Virulence Mechanisms of Common Uropathogens and Their Intracellular Localisation within Urothelial Cells

**DOI:** 10.3390/pathogens11080926

**Published:** 2022-08-17

**Authors:** Samantha Ognenovska, Chinmoy Mukerjee, Martina Sanderson-Smith, Kate H. Moore, Kylie J. Mansfield

**Affiliations:** 1Detrusor Muscle Laboratory, Department of Urogynaecology, University of New South Wales, St. George Hospital, Sydney, NSW 2217, Australia; 2Department of Microbiology, St. George Hospital, Sydney, NSW 2217, Australia; 3Illawarra Health and Medical Research Institute, Wollongong, NSW 2522, Australia; 4Molecular Horizons, University of Wollongong, Wollongong, NSW 2522, Australia; 5Graduate School of Medicine, University of Wollongong, Wollongong, NSW 2522, Australia

**Keywords:** uropathogens, overactive bladder, urinary tract infection, bacterial cystitis, invasion, virulence

## Abstract

A recurrent urinary tract infection (UTI) is a common debilitating condition whereby uropathogens are able to survive within the urinary tract. In this study, we aimed to determine if the common uropathogens *Escherichia coli*, *Enterococcus faecalis*, and Group B Streptococcus possessed virulence mechanisms that enable the invasion of urothelial cells. Urothelial cells were isolated from women with detrusor overactivity and recurrent UTIs; the intracellular localisation of the uropathogens was determined by confocal microscopy. Uropathogens were also isolated from women with acute UTIs and their intracellular localisation and virulence mechanisms were examined (yeast agglutination, biofilm formation, and haemolysis). Fluorescent staining and imaging of urothelial cells isolated from women with refractory detrusor overactivity and recurrent UTIs demonstrated that all three uropathogens were capable of intracellular colonisation. Similarly, the bacterial isolates from women with acute UTIs were also seen to intracellularly localise using an in vitro model. All Enterococcus and Streptococcus isolates possessed a haemolytic capacity and displayed a strong biofilm formation whilst yeast cell agglutination was unique to *Escherichia coli*. The expression of virulence mechanisms by these uropathogenic species was observed to correlate with successful urothelial cell invasion. Invasion into the bladder urothelium was seen to be a common characteristic of uropathogens, suggesting that bacterial reservoirs within the bladder contribute to the incidence of recurrent UTIs.

## 1. Introduction

Urinary tract infections (UTIs) are one of the most common types of infection found in clinical practice and are responsible for approximately 25% of all infections, affecting up to 60% of women at least once in their lifetime [1]. Unfortunately, many women experience recurrent episodes of UTIs, with 27% observed to have a second positive UTI within 6 months and up to 12.4% with a third episode within a 12-month period [2,3]. The incidence of a UTI increases with age and is also affected by a decline in health, increased morbidities, the place of residence (i.e., a nursing home), and a history of catheterisation and antibiotic use [3,4]. Post-menopausal women experience a greater frequency of UTIs, likely associated with decreasing oestrogen and subsequent changes in the vaginal pH [1,5,6].

Up to 80% of recurrent infections are observed to be caused by the same strain as the initial infection [1]. This occurrence has been attributed to the ability of uropathogens to localise within the host urothelial cells, evading antibiotic treatments and the immune system [7,8,9,10,11,12]. Uropathogenic *Escherichia coli* (UPEC), responsible for up to 90% of all UTIs, has been observed to possess a pathogenic cycle of urothelial cell invasion and intracellular replication before fluxing out into the extracellular environment to re-infect the host [7,8,9,13,14]. UPEC achieves this through its primary adhesion protein FimH [15,16], which binds to the mannose residues on uroplakin proteins expressed on the urothelial cell surface, triggering a signalling cascade and the internalisation of the bacteria [13,17,18]. UPEC also expresses other adhesion molecules, including P and S fimbriae [19]. Recent evidence has demonstrated that other uropathogens such as *Enterococcus faecalis* (*E. faecalis*) and Group B Streptococcus (GBS) possess the same potential to invade the host urothelial cells, although the mechanism responsible for the cell invasion is different. Several studies have reported that an *E. faecalis* invasion is dependent on biofilm production [20,21,22], whereas GBS is believed to internally localise by releasing a pore-forming toxin (β-haemolysin/cytolysin) to cause cellular injury and create a site of entry [12,23,24].

As a result of these virulence factors promoting uropathogen survival, recurrent episodes of UTIs may be associated with an increased incidence of the disease. Women with pre-existing medical conditions such as diabetes mellitus and conditions that necessitate prolonged catheterisation are found to be at a risk that a recurrent UTI may progress to pyelonephritis and urosepsis [25]. Another disorder where a recurrent UTI is considered to be of aetiological importance is detrusor overactivity (DO), wherein spontaneous or provoked detrusor muscle contractions are observed on the bladder filling [26], causing symptoms of urinary urgency, frequency, nocturia, and with or without urinary incontinence [27]. One third of DO patients do not respond to the first line therapy of anticholinergic medications and are deemed to be refractory with an unknown aetiology [28].

In the last decade, multiple studies have determined that 30–60% of patients with DO experience UTIs [29,30,31,32]. Additionally, elevated levels of inflammatory cytokines normally released during a UTI are found in the urine of women with urge incontinence [33,34,35,36]. Recent studies of the bladder microbiome have determined that women who failed to respond to an anticholinergic treatment (i.e., are refractory to the treatment) were likely to possess an increasingly diverse microbiome compared with healthy controls [37]. Overall, these results suggest that the presence of uropathogens within the bladder contributes to refractory DO.

This study was conducted in two parts. Part 1 aimed to determine whether common uropathogens UPEC, *E. faecalis*, and GBS were able to invade the urothelial cells of women with refractory DO and recurrent UTIs in vivo. This was conducted by undertaking confocal microscopy of urothelial cells isolated from urine specimens positive for the three uropathogens. Part 2 examined whether the uropathogens isolated from women with acute UTIs were capable of intracellular localisation in vitro. In addition, the presence of virulence mechanisms associated with intracellular localisation was observed. These included yeast cell agglutination, a measure of UPEC FimH adhesion protein activity [15,16]; biofilm production, associated with an invasion of *E. faecalis* [20,21,22]; and the haemolytic capacity, believed to be associated with a GBS-induced cellular injury [12,23,24].

## 2. Results

### 2.1. Part 1: Visualisation of Uropathogens In Vivo

Over the eight-month collection period, five MSU specimens were confirmed to be positive for UPEC, four specimens were positive for *E. faecalis*, and six were positive for GBS (Table 1). Confocal imaging and a subsequent Z-stack analysis (Figure 1) revealed the intracellular localisation of UPEC (Figure 1A–C). This localisation was confirmed in over three-quarters of the cells imaged from the *E. coli*-positive samples (10/13; Table 1). Similarly, *E. faecalis* (Figure 1D–F) was visually localised within the urothelial cell cytoplasm by confocal microscopy; this was also confirmed in over three-quarters of the cells imaged (10/13; Table 1). Unfortunately, imaging of urothelial cells from GBS positive samples was more difficult. A large proportion of the urothelial cells were ruptured and the slides were covered in cellular debris such that the location of the bacterium could not be accurately determined. A small number of intact cells (*n* = 7) were examined by confocal microscopy (Figure 1G–I) and the intracellular localisation of the bacteria was confirmed in just over 70% of these cells (5/7; Table 1). 

In 22 instances, polymicrobial growth (defined as the co-culture of ≥ 2 bacterial species in the absence of epithelial cells) was reported by the laboratory. Although this finding is typically regarded as contamination, a recent culture-independent PCR study revealed that samples with these cultures contained various uropathogens that could contribute to the occurrence of future UTIs [38]. As such, polymicrobial samples were included in the current study. These samples were fluorescently stained for the uropathogens of interest based on the previous MSU results reported for that patient (i.e., if a patient was previously positive for *E. coli*, then the MSU sample showing polymicrobial growth was stained using the anti-*E. coli* antibody). A total of 41 cells were imaged from the reported polymicrobial infections, of which 83% (34/41) were confirmed to contain intracellular bacteria (Table 1). This total cell number consisted of 25 cells stained for *E. coli*, wherein the intracellular localisation of bacteria was confirmed in 92% (23/25), and 16 cells were stained for *E. faecalis*, of which 69% were observed to contain intracellular bacteria (Table 1). Unfortunately, similar to the findings above, the polymicrobial samples stained for GBS were seen to contain cellular debris such that no clear images could be obtained.

### 2.2. Part 2: In Vitro Examination of Uropathogen Virulence Mechanisms

Uropathogen strains isolated from women with acute UTIs were examined for their ability to localise within RT4 urothelial cells (Table 2). The capacity of the bacterial isolates to localise within the urothelial cells varied even within the same uropathogen species. UTI89 and UTI89ΔfimH were used as positive and negative controls for the intracellular localisation of bacteria, respectively. Of the *E. coli* strains isolated from patients with acute UTIs, only EC1 and EC2 were able to localise within the RT4 cells. EC3 was seen to attach to the RT4 cells but remained extracellular. In contrast, all three of the *E. faecalis* isolates were observed to be localised within the RT4 cells. Two of the three GBS isolates (GBS2 and GBS3) appeared to be able to localise within the RT4 cells; however, GBS1 appeared adjacent to the RT4 cell membrane. 

An examination of three other potential virulence mechanisms was undertaken. These were chosen because previous studies have indicated that they may be related to the ability of uropathogens to localise within the urothelium. The positive and negative control UPEC strains behaved as expected; UTI89 immediately bound to the yeast cells to form aggregates (Table 2) whereas UTI89ΔfimH was unable to cause yeast cell agglutination. *E. coli* isolates EC1 and EC2 were also able to form aggregates, albeit more slowly. No yeast cell aggregation was observed with EC3 (Table 2). No *E. faecalis* or GBS isolates were observed to aggregate the yeast cells.

As expected, *E. faecalis* isolates were found to be the strongest producers of biofilms (Figure 2). GBS isolates also possessed an intermediate to strong capacity for biofilm production whereas the *E. coli* isolates were the weakest in terms of biofilm production (Figure 2). All three GBS isolates were seen to have a β-haemolytic capacity. Similarly, two of the *E. faecalis* isolates, EF1 and EF2, demonstrated β-haemolysis and EF3 demonstrated α-haemolysis. All UPEC isolates were seen to possess α-haemolytic capabilities.

## 3. Discussion

The major finding of this study was that common uropathogens—UPEC, *E. faecalis*, and GBS—were capable of localising within exfoliated urothelial cells from patients with DO and within a benign urothelial cell line. This finding is directly relevant to our understanding of how these bacteria survive within the urinary bladder to cause recurrent UTIs and also to the aetiology of refractory DO. A recent study revealed that inflammatory mediators associated with a bacterial infection lower the activation threshold of bladder afferent nerves, leading to hypersensitivity of the detrusor muscle with subsequent increased symptoms of urgency and frequency during bladder filling [39]. 

Immunofluorescence staining and confocal microscopy were conducted to accurately assess the location of these three bacterial species within the exfoliated urothelial cells of women with refractory DO who also experienced recurrent UTIs. The resulting Z-stack analysis confirmed all three uropathogen species to be residing within the urothelial cell cytoplasm. Similarly, the urothelial cells obtained from women with a diagnosis of a polymicrobial infection revealed both *E. coli* and *E. faecalis* to be present within the host cells, demonstrating that these uropathogens are able to maintain a consistent presence within the bladder. Whilst UPEC is well-known to intracellularly persist after invading urothelial cells [7,8,9,14], few studies have examined whether *E. faecalis* and GBS are capable of the same.

The invasive capability of uropathogens such as UPEC, *E. faecalis*, and GBS is likely to be a result of unique species virulence mechanisms; i.e., the expression of a D-mannose-binding phenotype [18], biofilm production [22], and the haemolytic/cytolytic capacity [12]. As such, this study sought to confirm the expression of these virulence mechanisms within strains isolated from women with UTIs and to determine whether these results correlated with the ability of the individual bacterial strain to invade the urothelial cells in vitro. Almost all isolated strains showed evidence of employing at least one of these virulence mechanisms.

The invasive capability of UPEC has been thoroughly documented to be driven by the primary adhesin protein FimH [15,16]. This protein binds to uroplakin-1a (UP1a) proteins expressed on the urothelial cell surface [40], subsequently leading to the internalisation of the bacterium [17,18]. A confocal image analysis of UPEC-infected RT4 cells revealed two of the three *E. coli* isolates (EC1 and EC2) were able to intracellularly localise. These same strains were also positive for yeast cell agglutination, indicating that the presence of FimH in these strains correlated with the invasive ability.

Although no studies have investigated the possible routes of *E. faecalis* invasion into urothelial cells, several studies have suggested that biofilm formation is the mediating factor of *E. faecalis* entry into other cell types [20,21,22]. All three *E. faecalis* isolates examined in the current study were found to be strong producers of biofilms in vitro, which correlated with the Z-stack analysis of the RT4 cultured cells that depicted each strain to have invaded the urothelial cells. Recent evidence suggests that this process may, in part, be due to increased expressions of Esp (enterococcal surface protein) and Sortase A (a membrane-bound enzyme), which may act to induce biofilm formation with cell wall anchoring and adherence to foreign materials/urothelial surfaces [41,42,43].

Similar to *E. faecalis*, GBS has been found to express Sortase A [44,45] and produce biofilms [46,47], which may aid in its invasive capacity [48]. Two of the three GBS isolates in the current study were confirmed to have localised within the RT4 cells and all three strains were observed to be producers of biofilms in vitro. Although no studies have examined the influence of GBS biofilms on intracellular localisation, it could be theorised that biofilm production had a degree of impact on the invasive ability of the strain.

One virulence factor that may aid in the host cell invasion of GBS is the pore-forming toxin β-haemolysin/cytolysin. GBS has been found to cause a cellular injury to create a site of entry into the urothelial cells [12] and other cell types [23,24]. In the GBS-positive samples, a large proportion of the exfoliated urothelial cells in the current study appeared to have been destroyed, suggesting that the GBS uropathogen has a high capacity for cellular lysis. The production of cytotoxic factors would explain such cellular destruction; large-scale lysis has been observed in two other studies [12,49]. The culture of the three isolated GBS strains onto HB agar demonstrated a haemolytic capacity although the co-culture of these strains with the RT4 cells did not cause a noticeable degree of cell lysis. It is, therefore, possible that another virulence mechanism, possibly biofilm formation, may have been responsible for the intracellular localisation of the GBS isolates in the current study.

Biofilm formation on catheters has been associated with UTIs [41]. It is interesting to note that two of the *E. faecalis* strains (EF2/EF3) were isolated from patients with short-term (< 8 days) suprapubic or indwelling urethral catheters. Biofilm formation is associated with the aggregation of bacterium and the secretion of extracellular polymeric substances onto the cell surfaces as a way of protection, adhesion, and additional co-aggregation [50,51]. This increases the survival of pathogenic bacteria as they can enter into symbiotic relationships with the surrounding bacteria and form communities [52,53]. In addition, the biofilms themselves provide an antibiotic resistance and shield the bacteria against the harsh extracellular environment [50,52]. In conditions such as refractory DO, this proliferation and the community formation of bacteria may be further exacerbated due to antibiotic therapies for recurrent UTIs. Not only would they induce the further formation of biofilms within multiple bacterial species [54], the use of antibiotics would also target healthy bacteria, skewing the natural balance of the bladder microbiome to one with a higher concentration of resistant uropathogens [55]. This could possibly explain the finding that women with refractory DO are likely to possess an increasingly diverse microbiome compared with healthy controls [37].

The results of this study showed that uropathogens associated with recurrent UTIs in women possessed the ability to localise within the urothelium. Fluorescent staining and imaging of the urothelial cells isolated from women with refractory DO and recurrent UTIs demonstrated that UPEC, *E. faecalis*, and GBS were all capable of intracellular colonisation. Similarly, the bacterial isolates obtained from women with acute UTIs were also seen to intracellularly localise using an in vitro model. The expression of the virulence mechanisms by these three uropathogenic species was observed to correlate with a successful urothelial cell invasion. An invasion into the bladder urothelium was seen to be a common characteristic of the uropathogens, suggesting that bacterial reservoirs within the bladder contribute to the incidence of recurrent UTIs.

## 4. Materials and Methods

### 4.1. Collection and Processing of Urine Samples

For Part 1 of this study, a series of mid-stream urine (MSU) samples was collected from four post-menopausal women (>50 years of age) with refractory DO. These women were recruited for this study as they had a history of recurrent UTIs with the uropathogens of interest. Ethics approval was obtained from the local human research ethics committee (Ethics approval number HREC14/193). The women were asked to be present for a review every few weeks over an eight-month period. At each visit, an MSU sample was collected (*n* = 39 in total). Each sample was split in half: one half was sent to the hospital microbiology laboratory for a routine culture (to a low count of >10^5^ colony-forming units (CFU)/L); the remaining half was processed to isolate the exfoliated urothelial cells. The samples were centrifuged (10 min, 160× *g*) and all but ~1 mL of the supernatant was removed. The pellet was resuspended with an equal volume of 10% Formalin (Sigma-Aldrich, St Louis, MO, USA) and subsequently cytospun (6 min at 120× *g*; Tharmac Cellspin I, ThermoFisher Scientific, Waltham, MA, USA) to allow the fixed urothelial cells to adhere to a glass slide.

### 4.2. Uropathogen Isolation and Culture

For Part 2 of this study, *E. coli, E. faecalis*, and GBS uropathogens were sourced from the hospital microbiology department. These uropathogens were isolated from the urine samples collected from female hospital patients with acute UTIs (age range: 29–94 years). In total, nine isolates were obtained, three of each uropathogen (Appendix A). The antibiotic profile and history of the presenting complaint were obtained for each isolate (Appendix A). The isolates were cultured and then stored in a solution of 10% glycerol. In addition, a well-characterised UPEC strain and a primary adhesion knockout variant, UTI89 [56] and UTI89ΔfimH [57], were examined.

All UPEC and *E. faecalis* strains were cultured on Luria-Bertani (LB) agar whereas GBS was cultured on horse blood (HB) agar; all were incubated at 37 °C overnight. The cultured isolates were subsequently stored at 4 °C. When required, a colony of each strain was suspended in an appropriate growth medium (Tryptone Soya Broth (TSB) for UPEC and *E. faecalis*; Todd Hewitt Broth (THB) for GBS), and statically incubated overnight at 37 °C.

### 4.3. Cell Culture and Invasion Assay

An RT4 cell line and transformed human urothelial cells derived from a benign papillary tumour of the bladder were obtained from the European Collection of Cell Cultures (ECACC). The RT4 cells were cultured in McCoy’s 5A culture medium (Gibco, available through ThermoFisher Scientific, Waltham, MA, USA) supplemented with 10% foetal bovine serum (FBS), 1% L-glutamine (L-Glut), and 1% penicillin/streptomycin (P/S) at 37 °C in 5% CO_2_. Once the cells reached a confluence, they were either passaged or seeded onto a sterilised glass coverslip (15 mm diameter) at approximately 0.2 × 10^6^ cells. The coverslips were placed in a Petri dish (35 mm diameter) and incubated for 30–60 min to allow the cells to adhere. The Petri dishes were subsequently flooded with antibiotic-free McCoy’s culture media and incubated overnight for the cells to achieve a 50–70% confluence.

For the invasion assay, each overnight bacterial culture was diluted (1:10) in fresh growth media and incubated until an optical density at 600 nm (OD600) of 0.5 was reached (60–120 min). The culture was then centrifuged (10 min, 5000× *g*), the supernatant was removed, and the bacterial pellet was resuspended in a 1:2 dilution of antibiotic-free McCoy’s culture media. The RT4 cells adherent to the coverslips were washed three times with phosphate-buffered saline (PBS) and then incubated with the resuspended bacteria for two hours to allow for the adherence and subsequent invasion of the urothelial cells. The coverslips were washed 3 times with PBS and incubated for a further 2 h with 100 µg/mL gentamycin to remove any extracellular adherent bacteria. The RT4 cells were washed again and fixed using 500 µL ethanol–acetic acid (95:5) for 15–20 min at −20 °C. Any remaining solution was removed, the cells were washed one final time, and the coverslips were stored at 4 °C until required. This invasion assay was performed in triplicate for each bacterial strain.

### 4.4. Immunofluorescence Staining and Confocal Imaging

The exfoliated urothelial cells adherent to the glass slides and the RT4 cells on the coverslips from the invasion assay were fluorescently stained to visualise the location of the bacterium with respect to the urothelial cell wall. The fixed cells were washed twice with Tris-buffered saline (TBS), and all UPEC and E. faecalis-positive samples were incubated with the conjugate antibody Wheat-Germ Agglutinin-Alexa Fluor-594 (WGA; 1:300; Thermo Fisher Scientific, Waltham, MA, USA) in PBS for 15 min to visualise the urothelial cell membrane. As WGA binds to the sialic acids of GBS, the AE1/AE3 protein was later targeted for this uropathogen only. The cells were washed twice and 10% Goat Serum (Sigma-Aldrich, St Louis, MO, USA) in PBS was applied for 30 min. The blocking serum was then removed and appropriate primary antibodies diluted in 0.1% Triton-X100 and 2% Goat Serum in TBS were applied: rabbit anti-*E. coli* (1:100; Meridian Life Sciences, Boca Raton, FL, USA); rabbit anti-Enterococcus (1:1000; Abcam, Melbourne, VIC, Australia); or rabbit anti-Streptococcus Group B (1:100; Abcam, Melbourne, VIC, Australia) with mouse anti-AE1/AE3 (1:100; Agilent Technologies, Santa Clara, CA USA). The slides were covered with aluminium foil and incubated overnight at 4 °C. After three washes, the secondary antibody of Goat anti-rabbit-Alexa Fluor-488 (1:200; Abcam, Melbourne, VIC, Australia) was added to all coverslips/slides. Samples positive for GBS were counterstained with Goat anti-mouse-Alexa Fluor-594 (1:200; Abcam, Melbourne, VIC, Australia) and the cells were further incubated for one hour. A final three washings were performed before using Prolong Gold Antifade Mountant with DAPI (Molecular Probes, Eugene, OR, USA). The cells were either coverslipped or placed onto slides.

Confocal imaging of the fluorescent slides was performed within one month of the staining using an Olympus FV1200 confocal microscope with Olympus FluoView software. Three or four images were taken of each sample with the pixel size resolutions optimised for the Alexa Fluor-488 fluorescence channel.

### 4.5. Yeast Cell Agglutination

The yeast strain *Saccharomyces cerevisiae* (*S. cerevisiae*) was obtained from Microbiologics Inc., Saint Cloud, MN, USA, and revitalised as per their instructions using Sabouraud Dextrose agar. A colony was subsequently cultured overnight in Yeast Extract-Peptone-D-Glucose (YPD) media at 37 °C and then measured for an OD600 of 0.5 alongside all bacterial isolates using their respective growth media. Equal volumes of yeast cells and each bacterial isolate were agitated within a 1.5 mL microtube. If aggregates formed within two minutes, the isolate was found to be positive for agglutination. UTI89 and UTI89ΔfimH served as a positive and negative control, respectively.

### 4.6. Biofilm Assay

All bacterial isolate cultures were diluted (1:20) in their respective media supplemented with 0.5% D-glucose (*w*/*v*). The bacterial solution was pipetted across two or three columns of a 96-well microtitre plate at 150 µL per well alongside control wells (TSB/THB–D-glucose). The plates were covered with AeraSeal film (Sigma, St Louis, MO, USA) and incubated with shaking (50 rpm) at 37 °C in air for 48 h, refreshing the wells with fresh media and D-glucose after the first 24 h.

The media and any non-adherent planktonic bacterial cells were carefully removed by a pipette and the wells were air-dried for 20 min. The adherent cells and biofilm (i.e., biomass) were stained using 150 µL of 0.2% (*w*/*v*) crystal violet (CV) and 1.9% ethanol and incubated at room temperature for 10 min. The solution was then removed and the wells were carefully washed twice with PBS to remove the unbound dye. The bound CV was solubilised using 150 µL of 1% SDS for 10 min at room temperature and then vigorously agitated. The solution was plated at a 1:5 dilution in 1% SDS and the absorbance read at 540 nm (A540). The assays were performed in triplicate for each bacterial isolate. The strength of the biofilm formation was categorised as: A540 < 0.5, non-biofilm-forming; 0.5–1.5, weak; 1.5–2.5, intermediate; 2.5–3.5, strong; and A540 > 3.5, very strong [58,59,60].

### 4.7. Haemolytic Assay

All bacterial isolates were plated onto HB agar and incubated overnight at 37 °C. The haemolytic activity was recorded according to the presence of a coloured zone around the bacterial colonies, which denoted the breakdown of red blood cells. A transparent yellow zone indicated a complete breakdown (β-haemolysis), a semi-transparent yellow–green zone indicated a partial breakdown (α-haemolysis), and no change in colour or opacity of the agar signified no haemolytic activity. This assay was repeated in triplicate with each successive attempt performed using a single colony from the previous cultured agar.

### 4.8. Data Analysis

Fiji Image J for Windows was used to analyse all confocal images for the purpose of visualising the bacterial localisation. The creation of all figures was performed using GraphPad Prism version 8.2.

## Figures and Tables

**Figure 1 pathogens-11-00926-f001:**
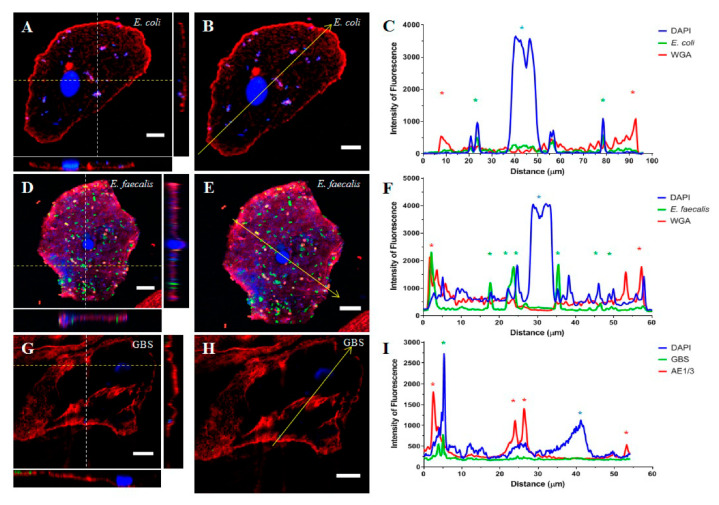
*E. coli* (**A**–**C**), *E. faecalis* (**D**–**F**), and GBS (**G**–**I**) invasion of an exfoliated urothelial cell from women with refractory DO. Cells were fluorescently stained and subsequently imaged using confocal microscopy. (**A**,**D**,**G**) Orthogonal views were obtained to visualise the location of all structures relative to the urothelial cell membrane. The white dashed line indicates the sampled YZ section whilst the yellow dashed line indicates the XZ section. (**B**,**E**,**H**) A line was drawn across a slice image occurring mid-way through the Z-stack. The intensity of fluorescence of each stain along the line was measured and depicted as a plot profile (**C**,**F**,**I**). The yellow arrow indicates the line and direction of the plot profile analysis whilst the coloured asterisks denote the objects the line passed through: entry and exit of the urothelial cell membrane (red), confirmed intracellular bacteria (green), and the urothelial cell nuclei (blue). Note, for *E. coli* (panel (**A**,**B**)) and GBS (panel (**G**,**H**)), the green staining showing intracellular bacteria was subdued. Although clearly visible when viewed in isolation, when all three channels were viewed together (as shown in the panels) the nuclear stain (blue) overwhelmed the green staining of the bacteria. Scale bars: 10 µm.

**Figure 2 pathogens-11-00926-f002:**
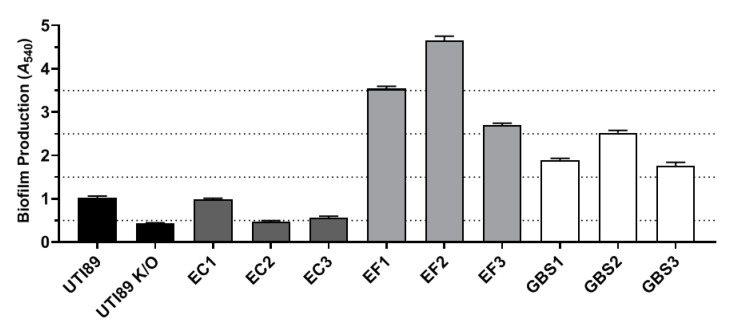
Biofilm production across all uropathogen isolates. An isolate with an average absorbance measurement of less than 0.5 is characterised as non-biofilm-forming; 0.5–1.5 is considered weak; 1.5–2.5 is intermediate; 2.5–3.5 is strong; and > 3.5 is a very strong biofilm producer. The dotted lines denote the thresholds between each of these categories. Data are displayed as mean ± SEM of three replicates.

**Table 1 pathogens-11-00926-t001:** Percentage of cells confirmed to contain intracellular bacteria for the individual uropathogen of interest for each MSU culture result.

MSU Result	Samples (*n*)	Confirmed Intracellular Localisation (%) *
*Escherichia coli*	5	76.9% (10/13)
*Enterococcus faecalis*	4	76.9% (10/13)
Group B Streptococcus	6	71.4% (5/7)
Polymicrobial	22	82.9% (34/41) *E. coli* (92.3%; 23/25) *E. faecalis* (68.8%; 11/16)

* Values in parentheses represent the number of cells confirmed to contain bacteria localised intracellularly using confocal imaging over the total number of cells imaged within that category.

**Table 2 pathogens-11-00926-t002:** Summary of virulence mechanisms possessed by each bacterial isolate.

Bacterial Isolate	Localisation within RT4 Cells	Yeast Cell Agglutination	Alpha-Haemolysis	Beta-Haemolysis	Biofilm Formation
UTI89	✓	✓	✓	–	Weak
UTI89ΔfimH	✗	✗	✓	–	Non-forming
*E. coli* 1	✓	✓	✓	–	Weak
*E. coli* 2	✓	✓	✓	–	Non-forming
*E. coli* 3	✗	✗	✓	–	Weak
*E. faecalis* 1	✓	✗	–	✓	Very strong
*E. faecalis* 2	✓	✗	–	✓	Very strong
*E. faecalis* 3	✓	✗	✓	–	Strong
GBS1	*	✗	–	✓	Intermediate
GBS2	✓	✗	–	✓	Strong
GBS3	✓	✗	–	✓	Intermediate

Ticks (✓) indicate a positive result for the corresponding test and crosses (✗) indicate a negative; dashes (–) represent a non-applicable result and an asterisk (*) represents an unconfirmed result.

## Data Availability

The data presented in this study are available on request from the corresponding author. The data are not publicly available due to privacy considerations.

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
