# Peer review of "Virulence Mechanisms of Common Uropathogens and Their Intracellular Localisation within Urothelial Cells"

_pathogens, 2022, doi:10.3390/pathogens11080926_

Round 1

Reviewer 1 Report

The work is interesting, but contains a small number of different experiments that would give a better insight into the issue under study. The authors forget that UPEC may also have other adhesion/invasion operons than the fim encoding type 1 fimbriae, eg adhesins from the Dr family which can also induce invasion.

In my opinion, the work should at least be supplemented with a genetic analysis of the tested strains regarding the presence of the most important virulence genes, e.g. by an ordinary PCR. Ideally, whole genomes should be sequenced, supplemented by additional transcriptome analysis.

The study of yeast agglutination ability should be supplemented with the ability of strains to agglutinate human erythrocytes in the presence of mannose and chloramphenicol (specific for Dr fimbria).

Supplementary materials should include photos showing the adherence patterns of the studied strains to cell lines, e.g. diffusion or aggregation type of adherence.

The standard biofilm method used is subject to a very large error. Very often, at the washing stage, the biofilm that does not show strong bond with PP is rinsed off. In supplementary materials, it would be worth presenting sequential photos collected over time showing biofilm formation and its final structure - biofilm cultivation statically on a 6-well plates under an inverted microscope with an environmental chamber.

In Figure 1 no bacteria are visible in panels A, B and G, H. It would be useful to give the average number of invaded bacteria per epithelial cell. It would also be worthwhile to specify the amount of bacteria in the appropriate urine samples.

The proposed additional experiments would significantly enhance the scientific impact of the research. However, the Reviewer does not consider them as a critical element necessary for the publication of the manuscript.

Reviewer 2 Report

I do not have many comments, this was a very interesting manuscript. The one thing I might expand on slightly is why did you include so few patients?  Were there limitations on the patient population or perhaps the tests you were doing that did not allow for more samples from other patients?  The results are very interesting, but it would also be interesting to see if these same results hold true for a larger population. 
